# Elucidation of the Epitranscriptomic RNA Modification Landscape of Chikungunya Virus

**DOI:** 10.3390/v16060945

**Published:** 2024-06-12

**Authors:** Belinda Baquero-Pérez, Enrico Bortoletto, Umberto Rosani, Anna Delgado-Tejedor, Rebeca Medina, Eva Maria Novoa, Paola Venier, Juana Díez

**Affiliations:** 1Molecular Virology Group, Department of Medicine and Life Sciences, Universitat Pompeu Fabra, Dr. Aiguader 88, 08003 Barcelona, Spain; 2Department of Biology, University of Padova, Via Ugo Bassi 58/b, 35131 Padova, Italy; enrico.bortoletto@phd.unipd.it (E.B.); umberto.rosani@unipd.it (U.R.); 3Center for Genomic Regulation (CRG), The Barcelona Institute of Science and Technology, Dr. Aiguader 88, 08003 Barcelona, Spain; anna.delgado@crg.eu (A.D.-T.); r.medina.amezcua@gmail.com (R.M.); eva.novoa@crg.eu (E.M.N.); 4Universitat Pompeu Fabra (UPF), 08003 Barcelona, Spain

**Keywords:** chikungunya virus, alphaviruses, RNA modifications, inosine, ADAR1, epitranscriptome

## Abstract

The genomes of positive-sense (+) single-stranded RNA (ssRNA) viruses are believed to be subjected to a wide range of RNA modifications. In this study, we focused on the chikungunya virus (CHIKV) as a model (+) ssRNA virus to study the landscape of viral RNA modification in infected human cells. Among the 32 distinct RNA modifications analysed by mass spectrometry, inosine was found enriched in the genomic CHIKV RNA. However, orthogonal validation by Illumina RNA-seq analyses did not identify any inosine modification along the CHIKV RNA genome. Moreover, CHIKV infection did not alter the expression of ADAR1 isoforms, the enzymes that catalyse the adenosine to inosine conversion. Together, this study highlights the importance of a multidisciplinary approach to assess the presence of RNA modifications in viral RNA genomes.

## 1. Introduction

The RNA components of the host translation machinery, including mRNAs, tRNAs, and rRNAs are post-transcriptionally modified by a plethora of chemical changes. The so-called epitranscriptome landscape comprises more than 170 different modifications [1,2]. For many decades, RNA modifications were considered as irreversible marks, and the most heavily modified RNA molecules (tRNA and rRNA) attracted the greatest attention. However, it was difficult to determine the location and abundance of modifications in both cellular and viral mRNAs because of their lower modification levels and the lack of techniques for tracing all possible mRNA modifications in a transcriptome-wide manner [3]. In 2012, a novel methodology enabled the transcriptome-wide mapping of the most common internal modification of cellular mRNAs, *N*^6^-methyladenosine (m^6^A) [4,5]. Such a breakthrough technology revealed that m^6^A is dynamically added by m^6^A-methyltransferases (writers) and removed by m^6^A-demethylases (erasers) [6]. Besides m^6^A, only 10 other RNA modifications have been described in cellular mRNAs, including inosine, *N*^1^-methyladenosine (m^1^A), 5-methylcytosine (m^5^C), *N*^7^-methylguanosine (m^7^G), and pseudouridine (Ψ) [7]. Currently, available methods can locate most of these mRNA modifications at single-nucleotide resolution in transcriptome data [8].

As for cellular mRNAs, it is widely believed that the genome of positive-sense (+) single-stranded RNA [(+) RNA] viruses can undergo different modifications. For example, liquid chromatography with tandem mass spectrometry (LC-MS/MS) analyses have identified complex and dynamic RNA modification patterns in the genome of diverse (+) RNA viruses during infection [9,10]. Although MS represents a useful tool to trace such changes, putative RNA modifications must be validated by using independent research approaches since the contaminant presence of highly modified cellular RNAs, such as tRNA or rRNA, even at very low levels, can lead to misleading interpretations of the resulting LC-MS/MS data. This is also the case for transcriptome-wide mapping methods of RNA modifications, especially the antibody-dependent ones, for which false positives have been widely reported [11,12,13,14]. The importance of multidisciplinary approaches to confirm the absence of RNA modifications in viral RNA has recently been highlighted for *N*^6^-methyladenosine (m^6^A) and *N*^1^-methyladenosine (m^1^A) [12,15]. Moreover, extensive controls are required, for example, an in vitro transcribed (IVT) viral RNA of the complete genome, devoid of modifications, is a crucial negative control for transcriptome-wide mapping methods [12]. Therefore, any modification signal detected in a specific viral RNA position must be interrogated with an orthogonal approach and discarded if the same signal is observed in the IVT control. For example, it was widely accepted that viral RNA transcripts from viruses that exclusively replicate in the cytoplasm were extensively m^6^A-modified. How this might occur despite the nuclear location of the m^6^A writing machinery was unknown. By applying such orthogonal approaches, we found no evidence of m^6^A modifications in CHIKV and DENV transcripts, two cytoplasmic-replicating viruses previously reported to be m^6^A-modified [16], demonstrating that m^6^A modifications are not a general trait of viral RNA genomes [12]. Here, we extended these analyses and carried out a detailed analysis of an array of RNA modifications in the CHIKV RNA genomes. CHIKV is a mosquito-transmitted virus that produces two 5′ capped and 3′ poly(A)-tailed (+) RNA species during replication. The genomic (gRNA) (11.8 kb) contains two open reading frames, the first of which is directly translated to produce the non-structural proteins. The second ORF is expressed from a subgenomic RNA (4.1 kb) transcribed during infection and is translated to produce structural proteins. We focus on CHIKV as a model system of a cytoplasmically replicating (+) ssRNA virus because it replicates at extremely high levels in human cells [12,17], a characteristic that greatly facilitates the required purification of genomic RNA (gRNA) and subgenomic (sgRNA) CHIKV RNAs for subsequent RNA modification analysis. By using LC-MS/MS, we first quantify the occurrence of 32 different RNA modifications in poly(A)-selected RNAs isolated from mock-infected or CHIKV-infected HEK 293T cells. The analysis suggested the potential presence of inosine (I) in CHIKV gRNA. However, neither RNA-seq analyses identified any A-to-I editing in the CHIKV gRNA, nor did CHIKV infection alter ADAR1 expression.

## 2. Materials and Methods

### 2.1. Cell Lines

HEK 293T (ATCC; CRL-11268; female) cells were cultured at 37 °C and 5% CO_2_ in Dulbecco’s modified Eagle’s medium with glutamine (DMEM) (Gibco, Paisley, UK) supplemented with 10% (*v*/*v*) fetal bovine serum (FBS) (Gibco, Paisley, UK) and 1% (*v*/*v*) non-essential amino acids (Gibco, Paisley, UK). BHK-21 Clone 13 (ATCC number CCL-10) were cultured in Glasgow Minimum essential medium (GMEM, Lonza, Slough, UK) supplemented with 10% (*v*/*v*) FBS and 10% (*v*/*v*) tryptose phosphate broth (BD Biosciences, Franklin Lakes, NJ, USA).

### 2.2. Viruses and Infection Conditions

Stocks of CHIKV strain LR2006-OPY1 (GenBank: DQ443544) were generated directly by infection of BHK-21 cells with a stock (gifted by Andres Merits, University of Tartu) previously obtained by electroporation of BHK-21 cells. CHIKV supernatant was then harvested at 24 h post-infection when clear cytopathic effects were observed. CHIKV stocks were titrated by standard plaque assay in HEK 293T cells. All CHIKV infections were carried out at an MOI of 4 and for a period of 8 h of infection unless otherwise stated. The viral inoculum was incubated for 1 h.

### 2.3. Poly(A)+ RNA-Selection and Isolation of gRNA and sgRNA

Total RNA was extracted using TRIzol (Thermo Fisher, Waltham, MA, USA) according to the supplier’s protocol, and a TURBO DNA-free Kit (Thermo Fisher, Waltham, MA, USA) was used to remove any contaminating DNA from RNA samples. Isolated RNA was further purified by ethanol precipitation, and 30–40 µg of total RNA was subjected to poly(A)-selection with the use of Dynabeads Oligo (dT)25 (Thermo Fisher, Waltham, MA, USA) according to the manufacturer’s instructions. Poly(A)+ RNA samples were concentrated before LC-MS/MS analysis with the use of an RNA clean and concentrator kit (Zymo Research, Irvine, CA, USA). For isolation of gRNA and sgRNA, poly(A)+ RNA was first passed through an RNA clean and concentrator kit (Zymo Research, Irvine, CA, USA), and then at least 5 μg of poly(A)+ RNA was separated in a native 1% (*w*/*v*) agarose gel (containing ethidium bromide), which was prepared and ran in 1× tris-acetate-EDTA (TAE) buffer. gRNA and sgRNA bands were then gel-extracted with the Zymoclean gel RNA recovery kit (Zymo Research, Irvine, CA, USA) following the manufacturer’s instructions.

### 2.4. Quantitation of RNA Modifications by LC-MS/MS Analysis

A total of 200 ng of poly(A)+ RNA isolated from HEK 293T cells and Huh7 cells and 300 ng each of gel-extracted gRNA and sgRNA were digested with a nucleoside digestion mix (New England Biolabs, Hitchin, UK) at 37 °C for one hour. Samples were prepared for LC-MS/MS analysis as previously described [12]. Samples were further desalted using HyperSep Hypercarb SPE Spin Tips (Thermo Fisher. Waltham, MA, USA) and run using an LTQ-Orbitrap XL mass spectrometer (Thermo Fisher Scientific, San Jose, CA, USA) coupled to an EASY-nLC 300 (Thermo Fisher Scientific (Proxeon), Odense, Denmark). The mass spectrometer was operated in positive ionisation mode with the nanospray voltage set at 2 kV and the source temperature at 200 °C. Full MS scans were set at 1 micro-scan with a resolution of 60,000 and a mass range of *m*/*z* 100–700 in the Orbitrap mass analyser. A list of masses was defined for further fragmentation (see Appendix A). Fragment ion spectra were produced via collision-induced dissociation (CID) at a normalised collision energy of 35%, and they were acquired in the ion trap mass analyser. The isolation window was set to 2.0 *m*/*z* with an activation time of 10 ms. All data were acquired with Xcalibur software v2.1. Serial dilutions were prepared using commercial pure ribonucleosides (see Appendix A for details) (1–2000 pg/µL) to establish the linear range of quantification and the limit of detection of each compound. A mix of commercial ribonucleosides was injected before and after each batch of samples to assess the instrument stability and to be used as an external standard to calibrate the retention time of each ribonucleoside. Acquired data were analysed with the Skyline software (v20.2.0.343) and extracted precursor areas of the ribonucleosides were used for quantification. Modified ribonucleosides were normalised by the median of the precursor area of AUGC. The raw MS data have been deposited to the MetaboLights repository [18] with the dataset identifier MTBLS8697.

### 2.5. Generation of In Vitro Transcribed (IVT) CHIKV RNA

Plasmid DNA encoding the CHIKV (pSP6-ICRES1) was linearised using FastDigest NotI (Thermo Fisher, Waltham, MA, USA). Linearised DNA was purified using phenol/chloroform/isoamyl alcohol (25:24:1 pH = 8.0) (Sigma-Aldrich, St. Louis, MO, USA) and used as a template for in vitro transcription with the mMessage mMachine SP6 kit (capped RNA; Thermo Fisher, Waltham, MA, USA). IVT reactions were incubated for 4 h at 37 °C in a volume of 20 μL, which included 2 μL of GTP. Following TURBO DNase I treatment (Thermo Fisher, Waltham, MA, USA), IVT RNAs were purified with the RNeasy mini kit (Qiagen, Hilden, Germany), and 0.5 μg was run on a denaturing formaldehyde gel to ensure integrity, while the remainder IVT RNA was stored at −80 °C.

### 2.6. IFN Pre-Treatment

HEK 293T cells were pre-treated for 24 h with 10,000 units/mL of human interferon alpha A (alpha 2a) (IFN-α-2a) (PBL assay science, Piscataway, NJ, USA).

### 2.7. siRNA Knockdown and ADAR1 Overexpression

A total of 0.15 × 10^6^ HEK 293T cells, seeded on 24-well plates the previous day, were transfected with either 100 nM of the specific ADAR1 Silencer Validated siRNA (siRNA ID: 119581, Thermo Fisher, Waltham, MA, USA) or 100 nM siGENOME non-targeting siRNA (Dharmacon, Lafayette, CO, USA) using 2 μL of lipofectamine 2000 (Thermo Fisher, Waltham, MA, USA) per transfection. At 6–8 h post-transfection, the media was changed. At 24 h post-transfection, cells were transfected again in the same way. When transfections were performed in a 12-well plate, these were performed in a similar manner as for 24-well plates, with the exception that the previous day 0.3 × 10^6^ cells were seeded and 4 μL of lipofectamine 2000 was used. When transfections were performed in a 6-well plate, 0.5 × 10^6^ cells were seeded, and 7 μL of lipofectamine 2000 was used. All CHIKV infections were performed at 48 h post-siRNA transfection. The infection outcome was measured via qPCR, Western blot, and standard plaque assay.

To ensure sufficient gRNA and sgRNA production in samples intended for deep-sequencing analysis, 1 × 10^6^ HEK 293T cells were initially seeded in a 6-well plate and subsequently transfected with siRNA as per the established protocol. For overexpression experiments, 1 × 10^6^ HEK 293T cells, seeded on 6-well plates the previous day, were transfected with 2.5 µg of plasmid DNA (pmGFP-ADAR1-p110) and 7 μL of lipofectamine 2000 per transfection. At 6–8 h post-transfection, the media was changed. For samples prepared for deep-sequencing analysis, CHIKV infection was performed at 24 h post-transfection. pmGFP-ADAR1-p110 was a gift from Kumiko Ui-Tei (Addgene, Watertown, MA, USA, plasmid # 117928).

### 2.8. Western Blotting

Cell pellets were lysed for 30 min in a lysis buffer (50 Mm Tris HCl pH 7.6, 150 mM NaCl, and 1% (*v*/*v*) NP-40) containing a complete protease inhibitor cocktail (Roche, Basel, Switzerland). Lysates were then centrifuged at maximum speed for 10 min, and the supernatant was saved. The protein concentration was quantified with a BCA protein assay kit (Thermo Fisher, Waltham, MA, USA), and lysates were then mixed 1:1 with 2× Laemmli buffer and boiled for 5 min at 95 °C. Lysates were then stored at −80 °C. Protein samples were run on SDS-PAGE gels and transferred onto nitrocellulose membrane (GE Healthcare, Chicago, IL, USA) via wet transfer for 1.5 h at 100 V. Membranes were blocked with TBS + 0.1% (*v*/*v*) Tween 20 (TBST) and 5% (*w*/*v*) dried skimmed milk powder for 1 h and then incubated overnight at 4 °C, with the relevant primary antibodies diluted in 5% (*w*/*v*) milk TBST, except for anti-phospho-eIF2α, which was diluted in 5% (*w*/*v*) bovine serum albumin (Sigma-Aldrich, St. Louis, MO, USA) in TBST. Membranes were washed 3× for 10 min with TBST and subsequently incubated for 1 h at room temperature with the appropriate horseradish peroxidase-conjugated secondary antibodies (GE Healthcare, Chicago, IL, USA) diluted (1:10,000) in 5% (*w*/*v*) milk TBST. Membranes were washed again 3× for 10 min and treated with SuperSignal West Femto Maximum Sensitivity Substrate (Thermo Fisher, Waltham, MA, USA), and chemiluminescence was acquired with a ChemiDoc MP imaging system (Bio-Rad, Watford, UK). Band quantification was performed with Image Lab Software, Version 6.1 (Bio-Rad, Watford, UK).

Antibodies used in Western blotting were rabbit anti-ADAR1 (D7E2M) (Cell Signaling Technology, Danvers, MA, USA), 14,175, 1:1000), mouse anti-β-actin (Sigma-Aldrich, St. Louis, MO, USA, A5441-.2ML, 1:5000), anti-CHIKV rabbit polyclonal antibodies capsid (1:5000), and nsP1 (1:5000), all gifts from Prof. Andres Merits (University of Tartu), as well as rabbit anti-PARP (Cell Signaling Technology, 9542, 1:1000), rabbit anti-eIF2α (Cell Signaling Technology, 9722S, 1:1000), and rabbit anti-phospho-eIF2α (Ser51) (Cell Signaling Technology, 9721S, 1:500).

### 2.9. Two-Step Quantitative Reverse Transcription PCR (qRT-PCR)

Total RNA from cells was extracted using TRIzol (Thermo Fisher, Waltham, MA, USA) according to the supplier’s protocol, and any contaminating DNA was removed with a TURBO DNA-free Kit (Thermo Fisher, Waltham, MA, USA). Reverse transcription reactions (20 μL) were carried out with SuperScript III reverse transcriptase (Thermo Fisher, Waltham, MA, USA) according to the manufacturer’s instructions and containing 4 µL 5× first-strand buffer, 1 μL of murine RNase inhibitor (New England Biolabs, Hitchin, UK), 1 μL of 10 mM dNTP mix (Thermo Fisher, Waltham, MA, USA), 50 ng of random hexamers (Bioline, Cincinnati, Ohio, USA), and 1 μg of total RNA (in a total volume of 11 μL). qPCR reactions (10 μL) consisted of 1× Power SYBR green PCR master mix (Thermo Fisher, Waltham, MA, USA), 0.5 μM of each primer, and 4.5 μL template cDNA. Cycling was performed in a QuantStudio 12K Flex (Thermo Fisher, Waltham, MA, USA). The cycling program included a hold stage of 50 °C for 2 min and 95 °C for 10 min, followed by 40 cycles of 95 °C for 15 s (denature step) and 60 °C for 1 min (anneal/extend step). After qPCR, a melting curve analysis was performed between 60 °C and 95 °C to confirm the amplification of a single product. The following primers were used: GAPDH Forward (Fw) TGTCAGTGGTGGACCTGAC, GAPDH Reverse (Rv) GTGGTCGTTGAGGGCAATG, NSP1 Fw AACCCCGTTCATGTACAATGC, NSP1 Rv GTACCTGCTCATCTGCCCAATT, ENV Fw AAGCTCCGCGTCCTTTACCAAG, ENV Rv CCAAATTGTCCTGGTCTTCCT, ADAR1-p150 Fw AATCCGCGGCAGGGGTATT, ADAR1-p150 Rv TGTGCTCATAGCCTTGAAATGG, ADAR1-p110 Fw GTGTCCCGAGGAAGTGCAA, and ADAR1-p110 Rv TGTCTGTGCTCATAGCCTTGAAA.

### 2.10. Cell Viability Assay

Determination of the cellular metabolic activity was performed using a CellTiter-Glo Luminescent Cell Viability Assay (Promega, Madison, WI, USA), according to the manufacturer’s manual. Luminescence was measured using an FB12 luminometer (Berthold detection systems, Bad Wildbad, Germany).

### 2.11. Library Preparation for RNA-Seq

RNA libraries were prepared using 50–100 ng of CHIKV in vitro transcribed RNA (IVT), purified gRNA, or sgRNAs). RNA libraries were prepared with the kit NEBNext Ultra II Directional RNA Library Prep Kit for Illumina (New England Biolabs, Hitchin, UK), following the section specified for “mRNA or rRNA Depleted RNA” with minor modifications. The final libraries were validated using a High Sensitivity D1000 Tapestation (Agilent, Santa Clara, CA, USA) and then pooled using an equimolar concentration for all the samples. The pool was quantified by qPCR and sequenced in a Miseq using a Nano Kit of 2 × 250 cycles. All the raw RNA-seq data have been deposited in the Sequence Read Archive (SRA) database from NCBI with the accession number PRJNA1096462.

### 2.12. ADAR1 Editing Analyses

The editing sites were traced using a dedicated pipeline. In detail, the reads were trimmed using trimmomatic [19] with the following parameters: “LEADING:3 TRAILING:3 MINLEN:50”, and FastQC (https://www.bioinformatics.babraham.ac.uk/projects/fastqc/, accessed on 15 February 2022) was used to assess the quality of the trimmed reads. The alignment of the trimmed reads on the CHIKV reference genome (NC_004162) was performed with the bwa mem algorithm [20] with standard parameters. Picard MarkDuplicates (http://broadinstitute.github.io/picard, accessed on 20 March 2022) was used to tag and remove the PCR duplicates. To increase the accuracy of the alignment, the lofreq viterbi command was used to compute a probabilistic realignment of the mapped reads. The putative editing sites were identified using three different tools: (i) lofreq call [21] (-- call-indels -- min-cov 5 -- max-depth 1,000,000 -- min-bq 30 -- min-alt-bq 30 -- min-mq 20 -- min-jq 0 -- min-alt-jq 0 -- def-alt-jq 0 -- sig 0.0005 -- bonf dynamic -- no-default-filter), (ii) REDItools [22] (-S-me 2-os 4-q 25-bq 35-mbp 15), and (iii) GATK HaplotypeCaller [23] with the default parameters. Only the editing sites traced by at least two of the different tools were further considered. Also, all the editing sites located in homopolymeric regions were excluded. All the format conversions were performed with SAMtools [24].

The hyper-editing tool [25] was applied after minimal modifications were required to overcome the software incompatibilities of the original version. The parameters were adapted by applying 5 for the minimum of edited sites at Ultra-Edit read (%); 60 for the minimum fraction of edit sites/mismatched sites (%); 25 for the minimum sequence quality for the counting editing event (PHRED); 60 for the maximum fraction of the same letter in the cluster (%); 20 for the minimum of the cluster length (%); and it was imposed that the hyper-editing clusters should not be completely included in the first or last 20% of the read.

### 2.13. Statistical Analysis and Chart Production

The Fisher exact test was computed in R 4.3.1 exploiting the rstatix function. The charts were produced using the ggplot2 [26], Gviz [27], GenomicFeatures [28], and pheatmap packages.

## 3. Results

### 3.1. Mass Spectrometry Analyses Reveal Inosine Enrichment in the CHIKV Genomic RNA but Not in the Subgenomic One

To address whether CHIKV RNAs carry RNA modifications, we first performed LC-MS/MS for the interrogation and quantification of 32 differently modified nucleosides (Appendix A). Poly(A)-selected RNA was isolated from CHIKV-infected HEK 293T and from CHIKV-infected Huh7 cells, both at 12 h post-infection, a time of high viral RNA production at which ≥50% of total mRNAs correspond to CHIKV RNAs [12,17]. Additionally, as a control, poly(A)-selected RNA was also isolated from the corresponding mock-infected HEK 293T and Huh7 cells. None of the detected modifications exhibited enrichment in CHIKV-infected cells compared to mock-infected cells; indeed, for some, modification levels were even lower (Figure 1). As shown before, consistent with m^6^A being the most common internal modification of cellular mRNAs, it was detected as the most abundant in both cell lines [12,17]. Inosine and 2′-O-methyladenosine (Am) were amongst the most abundant RNA modifications in both mock-infected and CHIKV-infected cells, whereas pseudouridine (Ψ) was only detected in Huh7 cells (Figure 1a). To distinguish between the RNA modification signals coming from viral and host RNAs, poly(A)+ RNA-selected samples from CHIKV-infected and mock-infected HEK 293T cells were run in native agarose gels, and viral genomic RNA (gRNA) and subgenomic RNA (sgRNA) were extracted from the corresponding bands (Figure 1b). Note that under native conditions, the gRNA segments migrate as two distinct bands, with the larger one likely corresponding to the double-stranded replication intermediate (i.e., the duplex of positive and negative strands). We avoided denaturing formaldehyde gels since they might lead to the artificial formation of *N*^6^-hydroxymethyladenosine (hm^6^A) [29]. LC-MS/MS analyses of the gel-purified CHIKV gRNA and sgRNA samples revealed a reduced number of modifications compared to the corresponding poly(A)+ RNA samples (Figure 1c), suggesting that the viral RNAs are not enriched in any of these modifications. m^6^A modification was the most abundant in the sgRNA samples and one of the most abundant in the gRNA samples; however, by combining both antibody-dependent and antibody-independent approaches, we have recently shown that this modification signal comes from contaminating host RNAs and not from viral RNAs [12]. While in this previous study, we focussed exclusively on the detection of m^6^A modification in CHIKV RNAs, in the present one, we inves-tigated the rest of the modifications to comprehensively elucidate the epitranscriptome of CHIKV RNAs. Ψ was absent from sgRNA samples but was detected in two out of three gRNA samples. As this modification is highly abundant in rRNA, the detected signal is much more likely to be a result of contaminating rRNAs (Figure 1c). Am was detected in both genomic and subgenomic CHIKV RNA but at lower levels in the latter. Again, this modification could arise from contaminating rRNAs. Intriguingly, inosine was detected in the three gRNA samples but absent from sgRNA samples (Figure 1c). Inosine is not present in rRNAs [30] and the proportion of contaminating cellular RNAs in the gRNA and sgRNA fractions is expected to be similar, suggesting that inosine could be enriched specifically in CHIKV gRNAs.

A-to-I editing is catalysed by the adenosine deaminases acting on RNA (ADAR proteins), which specifically deaminate adenosines in regions of double-stranded RNA. In mammals, there are three ADAR proteins: ADAR1, ADAR2, and ADAR3. ADAR1 is responsible for most of the A-to-I editing in cellular RNAs, including coding and non-coding RNAs, and it is critical in recognising and editing endogenous (self) and exogenous (viral) dsRNA duplexes to avoid triggering innate immune responses [29,30,31,32]. ADAR2 is predominantly expressed in the brain where it has a specific role in editing transcripts encoding neurotransmitter receptors [31,33]. Finally, ADAR3 is specifically expressed in the brain and lacks deaminase activity [30,32]. ADAR1 exists in two different isoforms, a constitutively expressed short isoform (ADAR1-p110) and a longer interferon-inducible isoform (ADAR1-p150). The former isoform is located mainly in the nucleus, while the latter is nuclear but also cytoplasmic [32], where it can interact with the viral dsRNAs that are recognised as non-self [32]. A-to-I editing can have three major consequences: (i) destabilisation of dsRNA structures, (ii) amino acid substitutions that alter the protein-coding potential, because inosine is read as guanosine by the host translation machinery, and (iii) generation or removal of splice sites as the host splicing machinery interpret inosine as guanosine. Because ADAR1-editing was reported in other positive-strand cytoplasmic-replicating RNA viruses such as hepatitis C virus (HCV) [32,34], we decided to focus our efforts on deciphering the interaction of CHIKV with the ADAR1-editing host cell’s machinery and on validating whether CHIKV RNAs undergo ADAR1-editing by orthogonal approaches.

### 3.2. CHIKV Infection Does Not Alter the Induction of ADAR1 Isoforms

Next, we assessed the expression of ADAR1 isoforms in HEK 293T cells that were either left untreated or treated with IFN-α-2a for 24 h. Cells were then either mock-infected or infected with CHIKV for 8 h. As expected, quantitative qPCR analysis revealed that the mRNA levels of the constitutively expressed ADAR1 isoform (ADAR1-p110) were not changed after IFN treatment, irrespective of the infection status of the cells. On the other hand, as expected, the mRNA levels of the IFN-inducible ADAR1 isoform (ADAR1-p150) were significantly increased in IFN-treated cells, particularly in mock-infected cells (Figure 2a). Intriguingly, mRNA levels of both ADAR1 isoforms did not fully correlate as both ADAR1-p110 and ADAR1-p150 protein levels were increased in IFN-treated cells, compared with untreated cells. These results agree with a previous study in which ADAR1-p110 was identified as an IFN-inducible protein that can originate from a leaky translation from the IFN-inducible ADAR1-p150 exon 1A mRNA [35]. Importantly, CHIKV infection did not alter the induction of either isoform compared with mock-infected cells. This is not unexpected since this cell line responds to IFN but its PAMP sensing pathway is impaired [36]. Despite the observed modest induction of ADAR1 protein isoforms after IFN-treatment the viral proteins nsP1 and capsid were strongly diminished in IFN-treated cells compared with untreated cells (Figure 2b). The IFN-inducible protein kinase R (PKR) was induced by IFN treatment as expected and used as a control protein.

### 3.3. ADAR1 Knockdown Inhibits CHIKV RNA and Protein Levels, and Virion Production

We then assessed the role of ADAR1 isoforms during CHIKV infection by performing ADAR1 depletion experiments using small interfering RNA (siRNA) in HEK 293T cells. Note that this siRNA targets exon 2 of ADAR1; thus, it knocks down both ADAR1 isoforms, ADAR1-p110 and ADAR1-p150. After 48 h of depletion, the viral proteins nsP1 and capsid were markedly reduced in depleted cells (Figure 3a). This reduction was accompanied by diminished RNA levels of gRNA (quantified with specific primers for NSP1) and sgRNA (quantified with specific primers for ENV) (Figure 3b). Moreover, depleted cells exhibited a ~70% reduction in virion production (Figure 3c). This anti-viral effect of ADAR1 knockdown was observed without cytotoxicity, demonstrated by a similar cell metabolic activity in control and depleted cells and a lack of activation of apoptosis (Appendix A).

### 3.4. ADAR1 Knockdown Leads to Phosphorylation of Eukaryotic Translation Initiation Factor 2α (eIF2α)

The observed inhibition of CHIKV infection upon ADAR1 depletion could be attributed to either a secondary effect associated with deficiencies in ADAR1 editing of host mRNAs that ultimately affect viral expression or to a direct effect of ADAR1 editing on CHIKV RNAs. Of note, ADAR1 silencing increased the p-eIF2α levels, irrespective of the infection status (Figure 4). As phosphorylation of eIF2α inhibits its essential role in the initiation of protein synthesis [37], the observed eIF2α phosphorylation is predicted to affect protein synthesis and thus viral production, as observed. The stable β-actin levels under conditions of eIF2α phosphorylation can be attributed to its inherent stability and the fact that, unlike viral proteins, it is synthesised prior to the onset of silencing. Similarly, a comparable observation was made regarding tubulin in the context of ADAR1 depletion and ZIKV infection [38].

### 3.5. RNA-Seq Analyses Detect Adenosine to Inosine (A-to-I) Editing in CHIKV RNAs

To validate these results with an orthogonal approach, we addressed the presence of inosine modification in CHIKV RNA by RNA-seq analyses. Genomic and subgenomic CHIKV RNAs were size-selected as in Figure 1b from CHIKV-infected HEK 293T cells under three experimental conditions: ADAR1-depletion (siADAR1), non-targeting siRNA (siControl), or ADAR-1-110 overexpression (OE_ADAR1). For each condition, three biological triplicates were generated. Based on the relative amounts of all possible 12 different single-nucleotide variants (SNVs), we computed an unsupervised clustering of 18 samples. A CHIKV in vitro transcribed (IVT) RNA was also deep-sequenced as a negative control. SNVs identified in the IVT samples were filtered out. The resulting heatmap of CHIKV RNA divided the samples into two distinct clusters (Figure 5A). The first cluster included all samples derived from siADAR1 and siControl experimental conditions without a clear-cut distinction between them. The second cluster included those from OE_ADAR1, with the genomic and subgenomic fractions grouped into two distinct subclusters. The OE_ADAR1 samples showed a clear enrichment in adenosine to guanosine variations but not in the other 11 variations analysed, as expected. Next, we computed the presence of edited AG sites along the CHIKV genome with a relative frequency higher than 0.1% in the different experimental groups and replicates (Figure 5B). A total of 26 of the 29 AG sites were identified in the OE_ADAR1 samples, while 3 appeared solely in the siRNA control or siADAR1 samples, but in general, in the OE samples, the presence of editing sites was more consistent between replicates. The observed editing differences between OE_ADAR1 samples and the other two sample groups were not related to differences in sequencing depth (Figure 5C, panels 2 and 4). Importantly, a Fisher test showed that only 7 AG sites were significantly edited, all belonging to the OE_ADAR1 samples, with 6 out of the 7 traced editing sites resulting in non-synonymous variations in the final protein (Figure 5C, panels 3 and 5). Nonetheless, the frequency of this editing was low, with 1% and 6% as minimum and maximum values, respectively. Thus, consistent with our previous Nanopore sequencing results, RNA-seq analyses did not identify any AG sites in CHIKV RNAs isolated from the siControl or siADAR1 samples. Solely when ADAR1 was overexpressed, a low level of AG editing was observed.

In agreement with these observations, we should note that previous nanopore direct RNA sequencing (DRS) analyses comparing in vitro transcribed CHIKV RNA (unmodified) and CHIKV native RNA (extracted from HEK293T-infected cells) also did not reveal any differences in their RNA modification patterns [12] using the NanoConsensus software [39]. Notably, NanoConsensus has been found to be robust in the detection of many different RNA modification types in DRS datasets when comparing two conditions (e.g., control vs native); however, inosine was not explicitly included in the benchmarking of NanoConsensus [39]. To ensure that *NanoConsensus* was able to identify inosine modifications, we re-analysed previously published DRS datasets that contained inosine-modified and unmodified oligonucleotides at known positions [40], showing that this modified inosine site can be accurately identified using NanoConsensus (Appendix A). Thus, nanopore DRS analyses provide additional support regarding the lack of A-to-I editing in CHIKV RNAs.

## 4. Discussion

In this study, we investigated the presence of 32 distinct modifications in CHIKV RNAs. Inosine was identified by MS as a putative modification in the gRNA (Figure 1c). However, further mapping of this modification at the base resolution by RNA sequencing revealed that it was not present in CHIKV RNAs (Figure 5). Thus, even though the viral RNA was the most predominant RNA in the gel-purified band, the inosine signal in the gRNA by LC-MS/MS was most likely attributed to residual contaminating cellular RNAs. Our results highlight the importance of orthogonal validations by base resolution sequencing methods to verify or refute the presence of putative RNA modifications.

Many viruses, both DNA and RNA, are believed to be subjected to RNA editing mediated by ADAR proteins [41]. Nevertheless, detailed investigations into A-to-I editing have been conducted for a few viruses, and the pro-viral or anti-viral functions of ADAR1 are mostly poorly characterised [41]. One of the first reports of inosine modification was described in hepatitis D virus (HDV), a circular ssRNA, where a single deamination of the adenosine occurs at the termination codon (UAG) of the small delta antigen [42], the only virus-coded protein. Because inosine is read by the translation machinery as guanosine, this deamination results in the translation of the codon as tryptophan (UGG) and the production of the large delta antigen, a 19 amino acid-long form of the small delta antigen, which is crucial for assembly of HDV [43]. This viral editing occurs in the nucleus and is carried out by ADAR1-p110 [44], a predominantly nuclear protein [45]. Another early study reported hyper-editing; i.e., multiple adenosines are converted to inosines in a short sequence region, in patients infected with the measles virus (MV) [46], a negative ssRNA virus. In very rare cases, MV can cause subacute sclerosing panencephalitis (SSPE), a severe persistent viral infection of the brain, which is typically fatal. Analysis of the viral genome isolated from patients with SSPE revealed several A-to-I substitutions in the measles matrix gene [46]. Because MV is a cytoplasmic-replicating virus, these editing events are thought to be carried out by ADAR1-p150, a protein that is both nuclear and cytoplasmic [45]. Hyper-editing of MV is believed to be proviral by promoting persistent infections, as the virus evades the host immune response by creating novel sequence variants [47]. Another virus subjected to inosine modification is the mouse polyomavirus (MPyV), which has a circular double-stranded DNA (dsDNA) genome. During late MPyV infection, the early RNA transcripts hybridize to complementary late RNA transcripts creating double-stranded RNA (dsRNA) structures that can be targeted by ADAR1. Analysis of early RNAs showed that many of these contained almost half of the adenosines changed to inosines or guanosines [48]. This hyper-editing is suggested to be responsible for the retention of modified early RNAs in the nucleus during late MPyV infection [47].

ADAR1 has been shown to enhance the replication of several RNA viruses, such as human immunodeficiency virus type 1, MV, vesicular stomatitis virus, and Zika virus, by inhibiting the antiviral PKR pathway in an RNA-editing-independent manner [38,49]. Interestingly, ADAR mutations drive PKR-linked immune diseases such as the Aicardi-Goutières syndrome [50,51]. Here, we observed a proviral role of ADAR1 in CHIKV infection (Figure 3), and ADAR1 depletion triggered phosphorylation of eIF2α, both in mock- and CHIKV-infected cells (Figure 4). These results are in line with previous studies using non-infected cells, in which overexpression of ADAR1-p150 or ADAR1-p110 decreased p-eIF2α protein levels in HEK 293T cells [52], and reciprocally, ADAR1 knockout A549 cells exhibited an increase in phosphorylation levels of eIF2α [38]. Consistently, overexpression of ADAR1-p150 or ADAR1-p110 decreased the levels of pPKR [52] and ADAR1 knockout increased them [38].

RNA sequencing analyses did not confirm the putative A-to-I modification in the CHIKV genome or transcripts, only forcing the overexpression of ADAR1-p110 isoform resulted in A-to-I editing. Moreover, we were not able to detect any hyper-edited sequences. In any case, we cannot eliminate the possible presence of inosine modifications at such low stoichiometry that they cannot be detected by RNA sequencing or nanopore sequencing analyses.

ADAR1-p110 overexpression has been shown to enhance replication of CHIKV and Venezuelan equine encephalitis virus (VEEV), another alphavirus, in STAT1−/− fibroblast cells [53], indicating a proviral role of ADAR in this context, potentially through modulation of dsRNA-dependent PKR-mediated stress responses known to inhibit CHIKV replication [41,54]. In these studies, the potential A-to-I editing of the viral genomes was not assessed [34]. In our study, we traced the change at the nucleotide level in CHIKV after overexpression of ADAR1-p110 and found several editing sites not traceable in a physiological situation, many of them resulting in a change of the protein-coding potential. For example, the sites located in position 7500 of the CHIKV genome would change the last non-structural protein, namely, the RNA-dependent RNA polymerase, producing a longer isoform that might have a different affinity with the RNA or a different RNA processivity.

In conclusion, our study highlights the need for a multidisciplinary validation of putative RNA modifications and underscores the importance of evaluating indirect effects when depleting RNA-modifying enzymes. With the ongoing progress in novel transcriptome-wide mapping technologies capable of achieving single-base resolution, such as direct RNA sequencing developed by Oxford Nanopore Technologies (ONT) [40,55], the elucidation of the viral epitranscriptome across various viruses can be systematically approached. These advancements promise to provide valuable insights into the complex landscape of RNA modifications and their roles in viral biology.

HEK 293T cells were transfected with siControl or siADAR1 for two consecutive days. A total of 48 h after the first transfection, cells were mock-infected or CHIKV-infected (MOI of 4) for 8 h. β-actin is shown as a loading control. β-actin-normalised values from depleted samples, below each band, are shown relative to their controls. Western blot quantification analyses are representative of two independent infections.

## Figures and Tables

**Figure 1 viruses-16-00945-f001:**
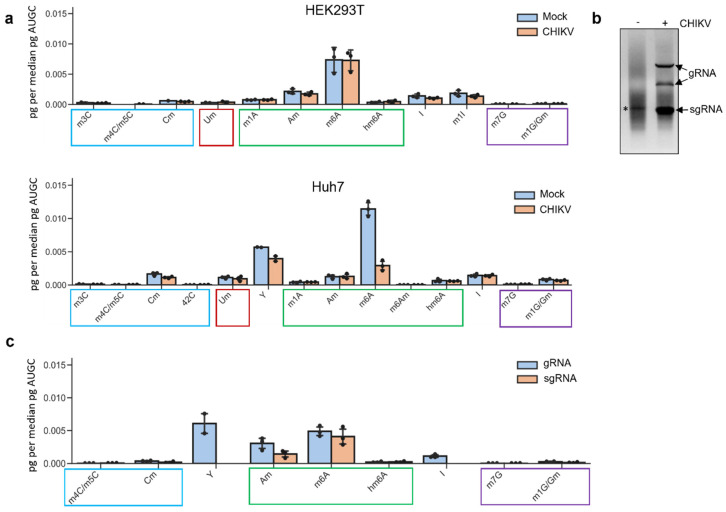
Analyses of the CHIKV RNA epitranscriptome by LC-MS/MS. (**a**) Cells were infected for 12 h p.i. with CHIKV at an MOI of 4, and poly(A)-selected RNAs were isolated from mock-infected or CHIKV-infected HEK 293T and Huh7 cells. A total of ~100 ng of digested ribonucleosides were analysed per sample for HEK 293T cells and ~75 ng for Huh7 cells. The bar chart shows mean values from three biological replicates with the error bars showing SD. Of the 32 RNA modifications screened, only those that were detected in at least two replicates are displayed. (**b**) A total of 5 μg of poly(A)+ RNA was isolated from mock-infected (−) or CHIKV-infected (+) (12 h p.i. MOI of 4) HEK 293T cells and then separated using a native 1% (*w*/*v*) agarose gel (containing ethidium bromide). The gel electrophoresis was carried out in 1× tris-acetate-EDTA (TAE) buffer. The asterisk indicates residual 28S rRNA. (**c**) HEK 293T cells were infected for 12 h p.i. with CHIKV at an MOI of 4. Poly(A)-selected RNAs were isolated from these and gRNA and sgRNA were further purified by gel-extraction. A total of ~150 ng of digested ribonucleosides were analysed per sample. The bar chart shows mean values from three biological replicates with the error bars showing SD. Of the 32 RNA modifications screened, only those that were detected in at least two replicates are displayed.

**Figure 2 viruses-16-00945-f002:**
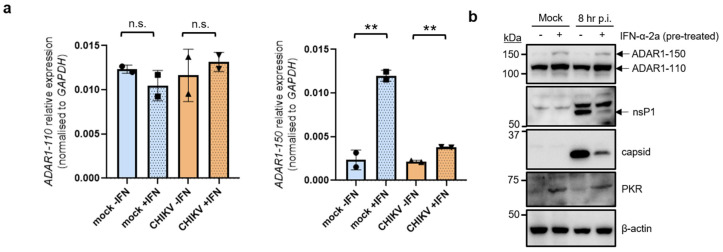
Protein expression levels of ADAR1 isoforms 150 and 110 are not affected by CHIKV infection. HEK 293T cells either were left untreated or treated with IFN-α-2a for 24 h. Cells were then either mock-infected or CHIKV-infected (MOI of 4) for 8 h. (**a**) qRT-PCR analyses to measure the expression of ADAR1 isoforms. The bar chart shows mean values from two biological replicates with the error bars showing SD. All statistical analyses were performed using a two-tailed *t*-test. n.s. = not significant, ** *p* < 0.01. (**b**) Western blot analyses of ADAR1 isoforms and viral proteins. β-actin is shown as a loading control. Western blots are representative of two independent infections.

**Figure 3 viruses-16-00945-f003:**
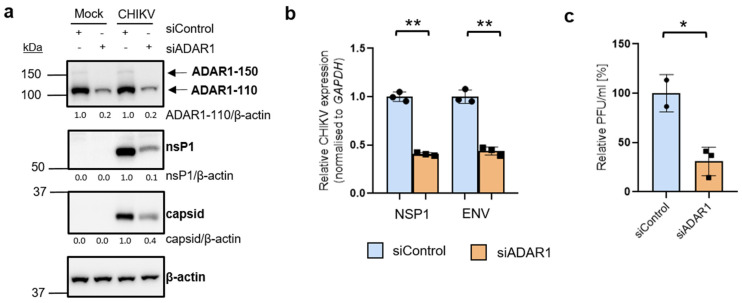
ADAR1 knockdown decreases viral RNA, viral protein levels, and viral titers. HEK 293T cells were transfected with siControl or siADAR1 for two consecutive days. A total of 48 h after the first transfection, cells were mock-infected or CHIKV-infected (MOI of 4) for 8 h. (**a**) β-actin is shown as a loading control. β-actin-normalised values from depleted samples, below each band, are shown relative to their controls. Western blot quantification analyses are representative of two independent infections. (**b**) Intracellular viral RNA levels were quantified by qRT-PCR and normalised against the housekeeping gene *GAPDH*. Viral RNA levels in depleted samples are shown relative to the corresponding controls. (**c**) Supernatants were collected at 8 h p.i. (MOI of 4) from CHIKV-infected control and knockdown cells were titered by plaque assay in HEK 293T cells. (**b**,**c**) The bar chart shows mean values from three independent infections with the error bars showing SD. All statistical analyses were performed using a two-tailed *t*-test. ** *p* < 0.01, * *p* < 0.05.

**Figure 4 viruses-16-00945-f004:**
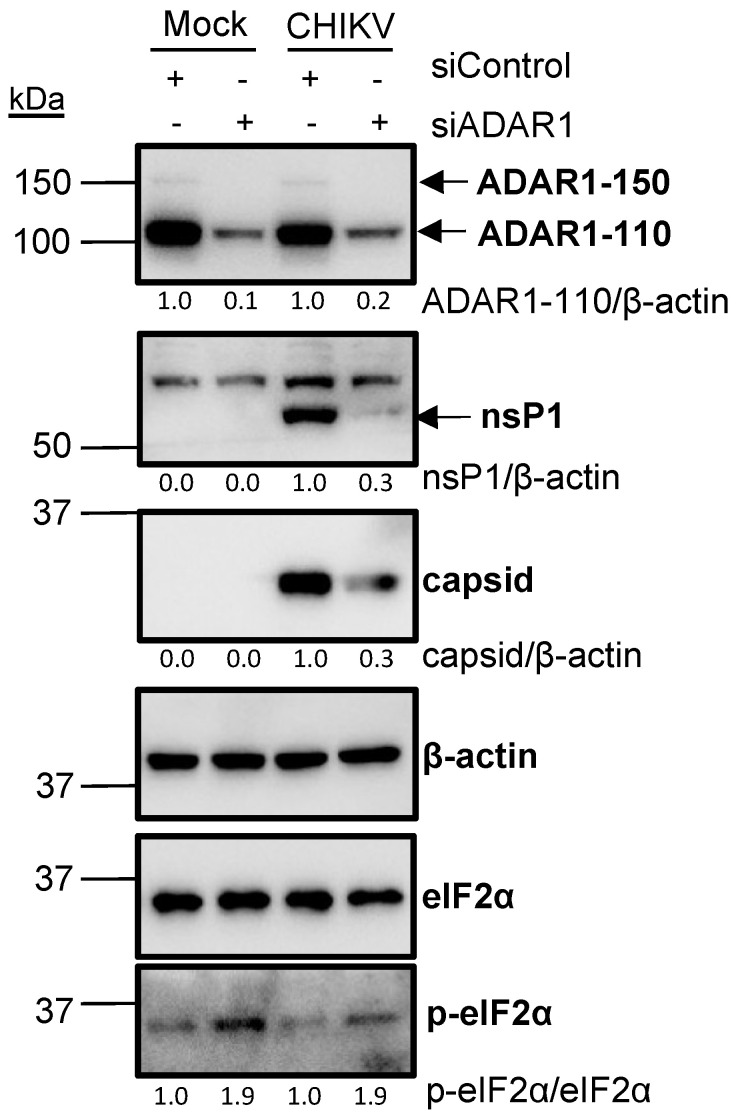
ADAR1 knockdown leads to phosphorylation of eukaryotic translation initiation factor 2α (eIF2α). HEK 293T cells were transfected with siControl or siADAR1 two consecutive days. 48 h after the first transfection, cells were mock-infected or CHIKV-infected (MOI of 4) for 8 h. β-actin is shown as a loading control. β-actin-normalised values from depleted samples, below each band, are shown relative to their controls. Western blot quantification analyses are representative of two independent infections.

**Figure 5 viruses-16-00945-f005:**
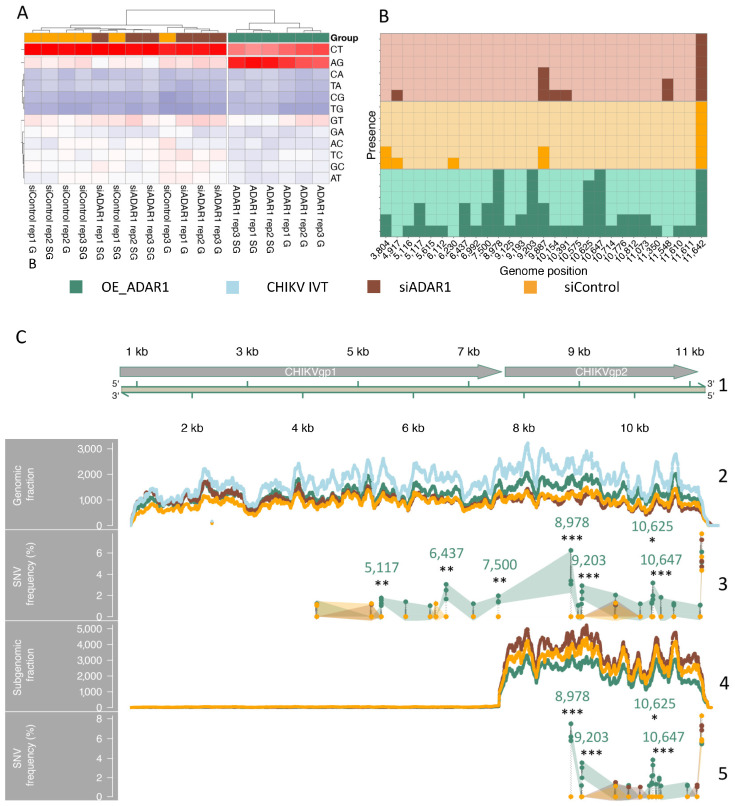
Inosine mapping by RNA-seq analyses. The three panels refer to CHIKV RNA reads from RNA-seq samples obtained from HEK 293T cells treated with specific ADAR1 Silencer siRNA (siADAR1, brown), non-targeting siRNA (siControl, yellow), ADAR1 overexpression vector (OE_ADAR1, green) or, as negative in vitro transcribed (IVT) control, from SP6-transcription of the chikungunya virus (CHIKV) genome (light blue). (**A**) Heatmap reporting the relative amount of all possible single nucleotide variations (SNV) in samples from the above-mentioned experimental groups. (**B**) The tile plot displays the occurrence of ADAR-compatible single nucleotide variants (SNVs) for each analysed sample (y-axis) across various positions in the CHIKV genome (x-axis), and each square along the y-axis represents one sample. Different groups are indicated by varying background colours, as explained above, and samples positive for editing sites are marked with a darker shade. (**C**) Representation of the sequencing depth and specific AG SNVs detected along the CHIKV genome. The different colours represent samples from different experimental conditions, as explained above. From the top to the bottom, arrows depict the viral transcripts CHIKVgp1 (non-structural polyprotein, NP_690588.1) and CHIKVgp2 (structural polyprotein, NP_690589.2) over the linear CHIKV genome track (1); the sequencing depth along the viral genome (2) or along the size-selected viral subgenome (4) are reported for each sample group (colours as reported above). Dots show the AG SNV frequency along the entire virus genome (3) or size-selected viral subgenome (5) (three samples per treatment group), whereas coloured areas indicate the confidence interval. Empty areas in (3) (0–3803 bp) and (5) (0–8977) indicate the absence of detectable AG SNV. The position and associated *p*-value symbol are reported (Fisher’s exact test, *p*-value * < 0.05, ** < 0.01, *** < 0.001).

## Data Availability

The raw mass spectrometry data have been deposited to the MetaboLights repository with the dataset MTBLS8697 (www.ebi.ac.uk/metabolights/MTBLS8697, accessed on 5 February 2024). The sequencing data from this study have been submitted to the NCBI SRA database under accession number PRJNA1096462. Raw fast5 files from inosine-modified and unmodified datasets that were used as a positive control for inosine detection using NanoConsensus were downloaded from ENA accession number PRJEB44511 (citar: https://www.nature.com/articles/s41467-021-27393-3, accessed on 5 February 2024).

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
