# Peer review of "Elucidation of the Epitranscriptomic RNA Modification Landscape of Chikungunya Virus"

_viruses, 2024, doi:10.3390/v16060945_

Round 1

Reviewer 1 Report

Comments and Suggestions for Authors

Here Diez and colleagues make a valuable contribution to research in epitranscriptomic modification and regulation of viruses. This work is of significant interest to RNA and virology fields. The authors present comprehensive mass spectroscopy analysis of CHIKV RNA which indicated inosine modification of genomic RNA species. Further orthogonal approaches failed to confirm this modification in unperturbed cells, though CHIKV RNAs were shown to be substrates for editing when ADAR1 was overexpressed. Interestingly, regulation of the virus by the A-to-I editing enzyme ADAR1 was identified using ADAR1 knockdown. A mechanistic clue was revealed by blotting for phosphorylation of translation factor eIF2-alpha. Though ADAR1 activity has been extensively linked to regulation of the dsRNA activated eIF2-alpha kinase PKR, the authors could not detect its activation in ADAR1 kd cells, though limitations of the antibodies used to assay this and other eIF2-alpha kinases was discussed. Overall the manuscript is well written, experiments well controlled (with an exception noted below) and with an exemplary and comprehensive methods section.

 Major points - 

                              Given the close relation of this study to the group’s preceding work on m6a modification of CHIKV RNA (ref 12), the introduction would benefit from a summary of those findings. 

                              It was unclear to me if the poly(A)+ MS data presented here is the same as that included in ref 12, but reanalysed or a separate experiment. Inclusion of a summary of the previous analyses as justification for this extended study/analysis would clarify this.

                              Line 408-410, it is stated prior nanopore analysis did not identify inosine in CHIKV RNA, though I could not find this described in ref 12. My understanding is nanopore base calling does not annotate inosines, was a specific analysis pipeline used? This could be included as a figure in this manuscript if it was not previously described in a prior publication, complementing the included RNA seq data (fig 5)

                              Attempts to identify the eIF2-alpha kinase responsible for enhanced eIF2-alpha phosphorylation in Fig 4 are problematic. Despite the obvious contender being PKR, the authors did not observe activated ph-PKR in their ADAR1 kd cells, though used an abcam antibody used extensively by others in published work (including my own lab’s). Without a positive control for detection of ph-PKR (eg transfection of poly(I:C)) however it is hard to draw any conclusions, and in my opinion blots for ph-PKR and ph-PERK should be omitted. Likewise, a control indicating the A92 treatment used was effective at inhibiting GCN2 was not included and so Fig 4b could also be omitted. These removals would not impact the significance of the manuscript’s findings. Alternatively, the authors may wish to identify phPKR and phPERK using the total protein antibodies and separating the modified forms by prolonged electrophoresis (we have done this successfully). To show that eIF2-alpha phosphorylation is responsible for impaired virus gene expression in ADAR kd cells they might also try rescuing this with ISRIB, which would nicely tie up this figure.

Minor errors and typos -

-          The impact of CHIKV on m6A in host mRNA was not commented on, if this was a separate experiment to that in reference to 12, repeated observation of this phenomenon is worth noting.

-          Line 300, I could not find in ref 12 where the statement that m6A signal detected by MS from viral RNA is shown to come from contaminating host RNA is supported. If this was simply previously concluded as orthogonal techniques did not support MS m6A detection that should be explained.

-          Line 340 – typo - levels repeated, perhaps the authors meant to say mRNA and protein levels do not correlate.

-          Line 347 – it would be clearer to specify that in 293Ts, PAMP sensing pathways are impaired/altered.

-          Line 424 – typo - variations

-          The w in western blotting does not need to be capitalized as it is not someone’s name

-          In the discussion ~line 567 the authors may wish to reference the role of PKR in the pathology of Aicardi-Goutières syndrome associated with ADAR1 mutations (eg PMID: 34343497) and cite the major publication identifying activation of PKR by endogenous dsRNA in ADAR1 ko cells (PMID: 34343497)

-          Line 589 – is 53 the correct reference here?

Author Response

 Major points - 

  1. Given the close relation of this study to the group’s preceding work on m6a modification of CHIKV RNA (ref 12), the introduction would benefit from a summary of those findings. 

We thank the reviewer for the suggestion and modified the introduction accordingly:

“... For example, it was widely accepted that viral RNA transcripts from viruses that exclusively replicate in the cytoplasm are extensively m6A-modified. How this might occur despite the nuclear location of the m6A writing machinery was unknown. By applying such orthogonal approaches, we found no evidence of m6A modifications in CHIKV and DENV transcripts, two cytoplasmic -replicating viruses previously reported to be m6A-modified, demonstrating that m6A modifications is not a general trait of viral RNA genomes”

  1. It was unclear to me if the poly(A)+ MS data presented here is the same as that included in ref 12, but reanalysed or a separate experiment. Inclusion of a summary of the previous analyses as justification for this extended study/analysis would clarify this.

The poly(A)+ MS data presented here is the same as that included in ref 12 but reanalyzed for additional modifications. Moreover, in the current study, quantification of m6A was shown as pg per median pg AUGC, while in the previous study it was represented as % m6A/A. A sentence has been added to the manuscript to clarify this point:

“..While in this previous study we focused exclusively on the detection of m6A modification in CHIKV RNAs, in the present one we investigated the rest of modifications to comprehensively elucidate the epitranscriptome of CHIKV RNAs,”

  1. Line 408-410, it is stated prior nanopore analysis did not identify inosine in CHIKV RNA, though I could not find this described in ref 12. My understanding is nanopore base calling does not annotate inosines, was a specific analysis pipeline used? This could be included as a figure in this manuscript if it was not previously described in a prior publication, complementing the included RNA seq data (fig 5)

The reviewer is correct that nanopore basecalling (“canonical” basecalling) does not annotate inosines. To detect RNA modifications (not only inosine), a specific pipeline was used, termed “Nanoconsensus” (Delgado-tejedor et al., bioRxiv 2023). This modification has been shown to detect multiple RNA modification types even at very low stoichiometries (Delgado-Tejedor et al., bioRxiv 2023). To identify RNA modifications in CHIKV using nanopore direct RNA sequencing, we compared native RNA molecules from CHIKV (potentially modified) with in vitro transcribed (IVT) CHIKV sequences (fully unmodified). Comparison of these two molecules should, in principle, capture any RNA modification that could be present in the CHIKV RNA molecule. This result was shown in our recent publication Baquero et al., Nat Comm 2024, where we focused the work on systematically assessing m6A modifications. We have now added a figure showing the comparison of IVT and native RNA CHIKV sequences using Nanoconsensus as new Supplementary Figure 2A (citing the previous work as source of data), and have also included a positive control to show that Nanoconsensus does identify inosine residues when comparing unmodified and modified RNA molecules, also included in the supplementary figure.

  1. Attempts to identify the eIF2-alpha kinase responsible for enhanced eIF2-alpha phosphorylation in Fig 4 are problematic. Despite the obvious contender being PKR, the authors did not observe activated ph-PKR in their ADAR1 kd cells, though used an abcam antibody used extensively by others in published work (including my own lab’s). Without a positive control for detection of ph-PKR (eg transfection of poly(I:C)) however it is hard to draw any conclusions, and in my opinion blots for ph-PKR and ph-PERK should be omitted. Likewise, a control indicating the A92 treatment used was effective at inhibiting GCN2 was not included and so Fig 4b could also be omitted. These removals would not impact the significance of the manuscript’s findings. Alternatively, the authors may wish to identify phPKR and phPERK using the total protein antibodies and separating the modified forms by prolonged electrophoresis (we have done this successfully). To show that eIF2-alpha phosphorylation is responsible for impaired virus gene expression in ADAR kd cells they might also try rescuing this with ISRIB, which would nicely tie up this figure.

 We agree with the reviewer and we have omitted the blots for ph-PKR and ph-PERK. The figure has been updated accordingly and the text modified.

Minor errors and typos -

1.The impact of CHIKV on m6A in host mRNA was not commented on, if this was a separate experiment to that in reference to 12, repeated observation of this phenomenon is worth noting.

The text has been changed to:  “..As shown before (12), consistent with m6A being the most common internal modification of cellular mRNAs, it was detected as the most abundant in both cell lines…. “

2.Line 300, I could not find in ref 12 where the statement that m6A signal detected by MS from viral RNA is shown to come from contaminating host RNA is supported. If this was simply previously concluded as orthogonal techniques did not support MS m6A detection that should be explained.

We have modified the text as follows to clarify this point:

“…however, we have shown by orthogonal techniques (both antibody-dependent and antibody-independent) that this modification signal comes from contaminating host RNAs and not from viral RNAs [12].

  1. Line 340 – typo - levels repeated, perhaps the authors meant to say mRNA and protein levels do not correlate.

We changed the text to “…Intriguingly, mRNA levels of both ADAR1 isoforms did not fully correlate as both ADAR1-p110 and ADAR1-p150 protein levels were increased in IFN-treated cells, compared with untreated cells.

4.Line 347 – it would be clearer to specify that in 293Ts, PAMP sensing pathways are impaired/altered.

The text has now been changed to “..This is not unexpected since this cell line responds to IFN but its PAMP sensing pathway is impaired”

4.Line 424 – typo – variations

The typo has been corrected

-The w in western blotting does not need to be capitalized as it is not someone’s name

The mistake has been corrected

- In the discussion ~line 567 the authors may wish to reference the role of PKR in the pathology of Aicardi-Goutières syndrome associated with ADAR1 mutations (eg PMID: 34343497) and cite the major publication identifying activation of PKR by endogenous dsRNA in ADAR1 ko cells (PMID: 34343497)

The references have been added

-  Line 589 – is 53 the correct reference here?

The reference was corrected.

We thank the reviewer for all the comments and suggestions.

Reviewer 2 Report

Comments and Suggestions for Authors

The manuscript titled Elucidation of the epitranscriptomic RNA modification land- 2 scape of chikungunya virusdescribes the study of RNA modifications in CHIKV-infected human cells. Mass spectrometry analysis identified 32 distinct RNA modifications showing enrichment of inosine in the genomic CHIKV RNA.

The manuscript is generally well-written and the results are logically presented. 

There are not many things that require revision, but for example, the font size in Figures 1 and 5 is too small and makes reading difficult.    

Comments on the Quality of English Language

Generally good, some fine tuning might be considered.

Author Response

The manuscript is generally well-written and the results are logically presented. 

There are not many things that require revision, but for example, the font size in Figures 1 and 5 is too small and makes reading difficult.    

  - The size was changed